# Transcription Factor *CcFoxO* Mediated the Transition from Summer Form to Winter Form in *Cacopsylla chinensis*

**DOI:** 10.3390/ijms25158545

**Published:** 2024-08-05

**Authors:** Chuchu Wei, Songdou Zhang

**Affiliations:** 1MOA Key Lab of Pest Monitoring and Green Management, Department of Entomology, College of Plant Protection, China Agricultural University, Beijing 100193, China; wxijia@outlook.com; 2Sanya Institute of China Agricultural University, Sanya 572025, China

**Keywords:** *Cacopsylla chinensis*, polyphenism, low temperature, *CcFoxO*

## Abstract

Amid global climate change featuring erratic temperature fluctuations, insects adapt via seasonal polyphenism, essential for population sustainability and reproductive success. *Cacopsylla chinensis*, influenced by environment variations, displays a distinct summer form and winter form distinguished by significant morphological variations. Previous studies have highlighted the role of temperature receptor *CcTPRM* in orchestrating the transition in response to 10 °C temperature. Nevertheless, the contribution of the transcription factor *FoxO* in this process has remained ambiguous. Here, we aimed to explore the correlation between *C. chinensis* FoxO (*CcFoxO*) and cold stress responses, while identifying potential energetic substances for monitoring physiological shifts during this transition from summer to winter form under cold stress by using RNAi. Initially, *CcFoxO* emerges as responsive to low temperatures (10 °C) and is regulated by *CcTRPM*. Subsequent investigations reveal that *CcFoxO* facilitates the accumulation of triglycerides and glycogen, thereby influencing the transition from summer form to winter form by affecting cuticle pigment content, cuticle chitin levels, and cuticle thickness. Thus, the knockdown of *CcFoxO* led to high mortality and failed transition. Overall, our findings demonstrate that *CcFoxO* governs seasonal polyphenism by regulating energy storage. These insights not only enhance our comprehension of *FoxO* functionality but also offer avenues for environmentally friendly management strategies for *C. chinensis*.

## 1. Introduction

The advent of global climate change has introduced formidable obstacles, marked by frequent occurrences of extreme weather events and escalating climatic unpredictability causing habitat alterations. Insects, with evolutionary origins dating back 350 million years or more to the Devonian period of the Paleozoic Era, have persevered through ice ages, warm interglacial periods, and now navigate the current era of climate change [1]. Insects that thrive in different habitats are considered “winners in evolution”, showing great adaptability. Polyphenism, a production of gene expression regulation, offers greater flexibility in response to complex environmental changes, enabling a single genotype to manifest multiple phenotypes under varied environmental stresses [2]. Prior researchers have partially unveiled the factors that regulate polyphenism. These factors include biotic elements like pheromones, along with abiotic elements such as population density, nutrition, photoperiod, and temperature [3,4]. Polyphenism shows how insects are smart in adapting to nature, working well with changes in the environment to survive, thereby bolstering the stability of insect populations crucial for their reproductive success [1]. Leveraging this mechanism of adaptive response, humans can better understand how to manage agricultural population under certain environmental conditions. For example, people can use RNA interference (RNAi) to prepare RNA biopesticides based on the key target genes of the adaptive process to control specific pests.

Despite the importance of polyphenism, contemporary investigations center on model insects. For example, in response to nutritional deficiencies pea aphids undergo wing development, transitioning from wingless to winged forms [5]. Early sources of nutrients can affect the caste differentiation in ants [6]. The underlying molecular mechanisms of these phenomenon are particularly fascinating. Recent discoveries have unveiled the involvement of neuroendocrine hormone signaling and transcription factors in their regulation. The insulin signaling pathway plays an important role in wing differentiation [7]. Insulin receptors assume a pivotal role in determining the differentiation of long and short wings in the brown planthopper [8], while FoxO exerts negative control over the development of wing morphology in pea aphids amidst varying environmental circumstances [5]. Moreove, the pathway also plays a role in wing polyphenism in *Pyrrhocoris apterus* [9]. Responding to population density, the dopamine signaling pathway regulates the switch of locusts from solitary to social behavior, with dopamine phosphorylation aiding in this shift [10,11]. In *Harpegnathos saltator*, kr-h1 serves as a key transcription factor responsive to hormonal signals like juvenile hormone (JH) and ecdysteroid (20E), binding to and regulating different target genes in worker ants and queen ants to establish their social hierarchy [12]. However, research on multiple transcription factors that ultimately regulate polyphenism in non-model species responses remains scarce and unclear.

*Cacopsylla chinensis* (Yang & Li), a member of Hemiptera: Psyllidae, exhibits a life cycle encompassing eggs, nymphs, and adults, posing a significant threat to pear orchards. In October 2011, 9.5 ha psyllid-infected orchards were defoliated in Japan [13]. With its piercing-sucking mouthpart, this insect induces leaf deformities and necrotic spots by extracting sap from pear leaves [14]. Additionally, it serves as a crucial vector for transmitting the economically detrimental *Erwinia amylovora* disease, which systematically attacks pear trees, leading to flower clusters dropping, fruits deforming, and branches withering, culminating in profound devastation [13]. Rather than entering diapause, *C. chinensis* survives the adverse environment through seasonal polyphenism, namely summer form and winter form [15]. Morphologically, the summer form displays a yellow–green hue and smaller stature, while the winter form presents as dark-brown and larger in size. As temperatures decrease, *C. chinensis* typically lays eggs along leaf veins before the leaf fall. The hatched nymphs exhibit traits suited for winter conditions (0–10 °C). Subsequently, with the temperature rise in early spring (15–25 °C), they lay eggs on rough bark, dormant buds, leaves, and flower buds, giving rise to summer-form nymphs [16]. Previous studies have identified the molecular mechanisms orchestrating phenotypic shifts in response to temperature variations. The third instar stage emerges as a critical juncture for the transition from summer form to winter form in *C. chinensis*. *CcTRPC3* acts as a cold receptor, sensing the low temperature around 10 °C and triggering the expression of cuticle-binding proteins *CcCPR4* and *CcCPR9*, eventually leading to alterations in total cuticle pigmentation, chitin content, and cuticle thickness [17]. Additionally, the low-temperature receptor *CcTRPM* influences the expression of rate-limiting enzymes in chitin biosynthesis, namely *CcTre1* and *CcCHS1*. The maintenance of the summer to winter transition is mediated by the stable expression of *CcTRPM* regulated by miR-252 [18]. Nevertheless, the precise regulation mechanism by downstream signaling, especially transcription factors, in response to temperature receptors remains elusive.

*FoxO* encodes a FOXO transcription factor belonging to the FOX superfamily’s “O” branch, featuring a highly conserved DNA-binding domain named the forkhead domain. The forkhead (FH) domain comprises three α-helices, three β-folds, a loop structure, and the C-terminus, crucial for DNA binding and specificity recognizing DNA sequences. Initially identified in human tumors, *FoxO* was recognized as a chromosomal translocation partner [19]. Mammals harbor four *FoxOs*-*FoxO1*, *FoxO3a*, *FoxO4*, and *FoxO6* located on distinct chromosomes, while invertebrates like *Caenorhabditis elegans* possess a single *FoxO* gene, *daf-16*, and *Drosophila melanogaster* houses *dFoxO* [20].

Researchers on *FoxO* in insects have primarily focused on regulating 20E and JH interactions during molting and metamorphosis, alongside attention to diapause, longevity, stress response, and other key physiological processes in insects [20,21]. In *Helicoverpa armigera* and *Bombyx mori* molting, 20E triggers increased FoxO expression and nuclear translocation, enhancing lipolysis in adipose cells [22,23]. In the developing ovary of *Harpegnathos saltator* queen ant, the insulin-resistant Imp-L2 reduces IIS-AKT/FOXO signaling pathway activity in the fat body, leading to dephosphorylation and nucleus translocation of FoxO, thereby activating longevity-associated genes [24]. In *lagomorphs*, FoxO aids in rapid responses to low temperatures, enabling survival under cold conditions by regulating gene expression associated with cold shock [25]. As a pivotal regulator downstream of the insulin signaling pathway, *FoxO* plays a crucial role in managing energy metabolism, influencing cellular energy balance, and closely relating to developmental processes and resistance to environmental stressors [26]. Consequently, *CcFoxO* might play a significant role in the shift from summer form to winter form in *C. chinensis*, which further elucidation is warranted.

To investigate the involvement of *CcFoxO* in the transition from summer form to winter form of *C. chinensis*, we utilized RNAi technology to disrupt the expression of *CcFoxO* and *CcTRPM*, studying the relationship between coldness and the function of *CcFoxO*. Phenotypic alterations were assessed by measuring total cuticle pigment content, cuticle chitin content, and cuticle thickness. Our study unveiled a significant role for *CcFoxO* in regulating the transition in response to low temperatures, impacting triglyceride and glycogen metabolism. This research aims to enhance understanding of the physiological function of the transcription factor *FoxO*, shed light on the ecological adaptation mechanism of *C. chinensis*, and potentially identify target genes for the effective control of psyllid pests.

## 2. Results

### 2.1. Cloing and Sequence Analysis of CcFoxO

The cDNA fragment, putatively encoding a forkhead box O protein, was identified as *CcFoxO* in *C. chinensis*. The full-length sequence of *CcFoxO* was confirmed through RT-PCR using specific primers (Appendix A) and subsequent sequencing. Sequence analysis revealed that the open reading frame (ORF) of *CcFoxO* spanned 1296 bp, encoding a polypeptide of 431 amino acids with a predicted molecular weight of 47.67 kDa and a pI of 4.80. Alignment analysis demonstrated a substantial amino acid identity of the FH domains of *CcFoxO* with FoxO sequences from four other selected insect species (Figure 1A). In an evolutionary perspective, *CcFoxO* exhibited the closest relationship with the DcFoxO homologue (*Diaphorina citri*, XP_026688498.1), both being significant Hemiptera pests of fruit trees (Figure 1B). The potential tertiary protein structure of CcFoxO was predicted using the online server Alphafold3 and refined with PyMOL-v1.3r1 software, revealing the presence of the conserved FH domain (Figure 1C).

### 2.2. Temporal and Spatial Expression Patterns of CcFoxO

The newly hatched summer-form nymphs were cultured at 10 °C and 25 °C, and the relative expression levels of *CcFoxO* were detected at 3, 6, and 10 days. Results from qRT-PCR revealed significantly higher expression levels at 10 °C (1.90, 2.83, and 4.77) compared to 25 °C (1.00, 1.05, and 1.82) (*p* values = 0.00138, 0.00419, and 0.00092) (Figure 2A). The differential expression under the two temperature conditions indicated that low-temperature exposure significantly stimulated the expression of *CcFoxO*.

Under the 10 °C condition, knockdown of *CcTRPM* led to a notable decrease in the relative expression of *CcTRPM*, demonstrating the high efficiency of the RNAi experimental system (Figure 2B). Intriguingly, the downregulation of *CcTRPM* expression also considerably reduced the relative expression of *CcFoxO* (Figure 2C). The observed positive correlation between *CcFoxO* and *CcTRPM* expression manifests that these two genes are interconnected within the same signaling pathway.

### 2.3. Impact of CcFoxO on Triglyceride and Glycogen Levels in C. chinensis

The newly hatched summer-form nymphs were subjected to 10 °C conditions and fed with dsEGFP in the control group and dsCcFoxO in the experimental group. After a 15-day treatment period, triglyceride and glycogen levels were assessed. Each treatment consisted of nine biological replicates, with 30 nymphs per replicate. Analysis showed a significant reduction in triglyceride and glycogen levels following dsCcFoxO treatment compared to nymphs treated with dsEGFP (Figure 3A,B). Additionally, Nile red staining demonstrated a marked decrease in lipid droplet abundance post-interference with *CcFoxO* expression based on both the lightness and size (Figure 3C).

These discoveries suggest that *CcFoxO* plays a crucial role in maintaining the abundance of triglycerides and glycogen, essential substances representing the primary forms of lipids and carbohydrates in insects. These compounds serve as essential nutrient reserves, indicating that *CcFoxO* is involved in nutrient metabolism under cold temperature.

### 2.4. Involvement of CcFoxO in Regulating Total Cuticle Pigment, Cuticle Chitin Content, and Cuticle Thickness of C. chinensis

Following a 15-day dsCcFoxO treatment of first-instar nymphs, the total cuticle pigment content was significantly lower compared to the dsEGFP treatment group. The visible color difference in the pigment extract was noticeable to the naked eye (Figure 4A). Under disrupted *CcFoxO* expression conditions, first-instar summer-form nymphs were maintained at 10 °C. After 15 days, the cuticle chitin content (1.00) and thickness (3.22 μm) were significantly reduced than those treated with dsEGFP (0.30, 1.62 μm) (Figure 4B,C). Significant differences in cuticle thickness were also observed in the representative TEM image (Figure 4D).

These results indicated that disruption of *CcFoxO* affects *C. chinensis*’s ability to adapt to low temperatures in terms of cuticle composition and thickness, highlighting the role of *CcFoxO* in epidermal alterations.

### 2.5. Role of CcFoxO in Modulating the Transition from Summer Form to Winter Form

At 10 °C, the results of nymphs treated with dsEGFP or dsCcFoxO are divided into two categories: successful transformation or death. The transition percent of summer-form first instar nymphs to winter-form nymphs significantly decreased after 15 days of dsCcFoxO feeding (from approximate 82.85% to 29.09%) (*p* value = 1.89282 × 10^−10^) (Figure 5A). Moreover, the mortality under this treatment was significantly higher than that of the dsEGFP treatment group (from approximate 17.15% to 70.91%) (Figure 5B). Notably, nymphs in the dsFoxO-treated group exhibited light-colored, yellow–green, and small traits after the 15-day period, while those in the dsEGFP-treated group displayed dark, black–brown, and slightly larger characteristics (Figure 5C). These observations suggest that *CcFoxO* is involved in the transition from the summer form to winter form of *C. chinensis*, actively facilitating this transition under low-temperature conditions.

## 3. Discussion

The phenomenon of polyphenism is widespread in the insect realm, with current research predominantly focusing on model insects like butterflies, moths, aphids, locusts, ants, and bees [27]. Among these, the mechanisms of wing dimorphism in aphids stands out as a well-studied example following locusts’ morphological transformations [28,29]. These reported phenomena are primarily influenced by population density, alongside factors such as nutrition, temperature, and photoperiod [30]. The shift from a solitary phase to gregarious phase in locusts is governed by a biogenic amine signaling pathway [31]. Polyphenism, which entails the expression of different phenotypes from the same genetic background under different biotic or abiotic stresses, plays a pivotal role in the survival and reproduction of insects [2]. Some insects exhibit seasonal polyphenism, exemplified by cases such as *Nemoria arizonaria*, where caterpillar closely mimic the seasonal morphology of the host plant [32], and *Bicyclus anynana*, where the size of hindwing eyepatches changes with the seasons [33]. The seasonal polyphenism of these two insects may help them adapt to temperature changes along with season on the one hand, and also help them disguise themselves to avoid being hunted by natural enemies on the other hand. Similarly, the subject of this study, *C. chinensis*, demonstrates seasonal polyphenism by responding to 10 °C and transitioning from summer-form to winter-form phenotypes [18]. Cuticle thickness, cuticle chitin content, and cuticle pigment content, regulated by bursicon, serve as discerning indicators for phenotypic variations between summer and winter variants [34].

In this study, expression analysis of *CcFoxO* revealed an upregulation in its expression in response to low temperature. Additionally, the expression patterns of *CcFoxO* and *CcTRPM* were consistent under low-temperature conditions based on our previous publication [18]. *CcTRPM*, identified as a temperature receptor in *C. chinensis*, responds to low temperatures, regulates chitin synthesis, influences cuticle characteristics, and aids in reducing heat loss in cold temperatures [18]. Thicker skin and darker body color help psyllids better reduce water loss and protect body temperature in the cold winter to adapt to the environment. Treatment with dsCcFoxO hindered the changes in cuticle properties and total pigment content, resulting in a significant decrease in the transition percent from summer form to winter form and an increase in mortality. Based on these findings, it is reasonable to infer that *CcFoxO* acts downstream of *CcTRPM*, linking these two genes and mediating insect physiological processes in *C. chinensis* triggered by low temperatures. Insects such as ectotherms experience fluctuations in body temperature according to the surrounding environment. To combat challenges like cold stress and food scarcity during winter, many species accumulate fat reserves beforehand. Fat serves as an energy reservoir during colder months and significantly affects reproductive capacity [35]. Notably, disruption of *CcFoxO* expression led to reduced triglycerides and glycogen content, accompanied by a partial absence of lipid droplets. This highlights the involvement of *CcFoxO* in the transition from the summer form to winter form of *C. chinensis*, affecting energy reserves, cuticle chitin biosynthesis, and cuticle properties, aligning with previous findings. For instance, *Aedes aegypti* increases triacylglycerol accumulation in dormant eggs to cope with cold environments as it migrates from tropical to temperate regions. The species adjusts energy reserves by increasing proteins and carbohydrates when triacylglycerol levels are low in a given population [36]. Similarly, insects enhance cold tolerance by increasing glycogen reserves during cold stasis. Glycogen broken down into sugar alcohols and protective compounds aids in surviving low temperatures [37]. Therefore, the elevation in triglycerides and glycogen, regulated by *CcFoxO*, represents an adaptation to cold conditions during the winter-form stage. Under dsCcFoxO treatment, the regulating of glycogen and triglycerides may be the first and then decrease in the transition percent.

Previous researchers have recognized the *FoxO* family gene as a forkhead transcription factor present in both mammals and insects. Following dsCcFoxO treatment, *C. chinensis* failed to exhibit corresponding phenotypic changes in response to low temperatures, showing no alterations in cuticle properties or pigment content. *FoxO* is known to be involved in the PI3K/Akt signaling pathway and the MAPK signaling pathway, impacting *FoxO* activity through phosphorylation and nuclear localization. The PI3K/Akt pathway is vital for cell growth, metabolism, and survival, responding to insulin and growth factors [38]. The MAPK pathway triggers FOXO phosphorylation in response to stressors, affecting cellular responses to oxidative stress, nutrient deprivation, and other external stimuli [39]. Both pathways influence cellular metabolism, including the metabolism of energy substrates like glycogen and triglycerides. The involvement of *CcFoxO* in these signal pathways also may be the reason for high mortality in dsCcFoxO treatment. While our study confirms the role of *CcFoxO* in regulating glycogen and triglycerides in response to low temperatures, the upstream regulatory components remain ambiguous and warrant further exploration. Further studies could explore the involvement of specific signaling pathways by monitoring changes in associated indicators.

Unlike prior research focusing on *FoxO*’s role in modulating ecdysteroids and juvenile hormones during insect molting, this study sheds light on *FoxO*’s reaction to seasonal polyphenism for the first time. Short-daylight and low-temperature conditions elevate *FoxO* activity in the moth *H. armigera*, reducing TGFβ signaling and inducing pupal diapause [40]. Both diapause and seasonal polyphenism exhibit protective mechanisms against low temperatures, underscoring FoxO’s involvement in responding to cold conditions in both phenomena. Studies on *FoxO* can enhance our understanding of its functional roles and elucidate the correlation between diapause and seasonal polyphenism by investigating the governing signaling pathways. As shown in Figure 6, at 10 °C, the low-temperature receptor *CcTRPM* was activated in first instar summer-form nymphs leading to increased expression of *CcFoxO*, facilitating energy accumulation, promoting cuticle chitin synthesis, and increasing cuticle thickness. Subsequently, the nymphs transitioned into the third instar winter form, enhancing adaptability to low temperatures and reducing mortality rates. These findings not only deepen our understanding of *FoxO*’s functionality but also provide new insights for environmentally friendly prevention and the control of *C. chinensis*.

## 4. Materials and Methods

### 4.1. Host Plant and Insect Rearing

The summer form and winter form of *C. chinensis* were collected from the “Lixiangzhuangyuan” pear orchards in Daxing District, Beijing City, China, during the summer and later fall seasons, respectively. The wild-type populations were purified through more than 20 generations of inbreeding in the integrated pest management (IPM) laboratory at China Agricultural University. Nymphs and adults from both forms were reared on pear seedlings enclosed with dense mesh screens. The summer form was reared at 25 ± 1 °C and 65 ± 5% relative humidity, while the winter-form was reared at 10 ± 1 °C and 25 ± 5% relative humidity, both under a 12 L: 12 D photoperiod. The host plants, 2–3-year old Korla fragrant pear seedlings ranging from 60–100 cm in height, were propagated from cuttings and maintained with regular watering and fertilization.

### 4.2. Molecular Characterization of CcFoxO

The FoxO homolog was identified from the transcriptome database of *C. chinensis*, and the accurately sequenced fragment was designated as *CcFoxO* (GenBank accession number: PP931994). The SMART online tool (http://smart.embl-heidelberg.de/, accessed on 25 June 2024) was used for predicting conserved domains. Subsequently, the three-dimensional structure of the protein was predicted using Alphafold 3 (https://alphafoldserver.com/, accessed on 25 June 2024), with further modifications performed using PyMOL-v1.3r1 software. Homologous protein sequences of CcFoxO were retrieved from NCBI BLASTP (https://blast.ncbi.nlm.nih.gov/, accessed on 25 June 2024) for 3 Lepidoptera, 3 Hemiptera, 2 Hymenoptera, and 2 Coleoptera insect species. The selected protein sequences exhibit lower E values and higher homology across various species in the search results, encompassing both closely and distantly related organisms. This approach enhances the clarity and accuracy of phylogenetic tree presentation. Amino acid sequence alignment analysis of the 12 protein sequences mentioned above was performed using DNAman 9.0 software. Phylogenetic analysis and the construction of a phylogenetic tree were carried out using the neighbor-joining (NJ) method in MEGA10.1.8 software.

### 4.3. qRT-PCR Analysis for mRNAs

To determine the mRNA expression levels of *CcFoxO* under different temperature conditions, summer-form 1st instar nymphs were gathered at 3, 6, and 10 days post-exposure to temperature of 25 and 10 °C in the incubators. Subsequently, these nymphs were promptly preserved in an ultra-low temperature freezer (−80 °C) for total RNA extraction.

Total RNA extraction for mRNA from the above samples was performed, employing the TaKaRa RNA Extraction kit (Cat# 9767, TaKaRa, Shiga, Japan) based on the kit protocol. Subsequently, the synthesis of the first-strand cDNAs was carried out using the Takara PrimeScript RT reagent kit (cat. no. RR047A, Takara, Shiga, Japan) following the manufacturer’s guidelines. The quantification of mRNA was executed utilizing the Bio-Rad CFX Connect Real-Time PCR System (Bio-Rad, Hercules, CA, USA) with the Takara TB Green Premix Ex TaqII kit (catalog no. RR820A, Takara, Shiga, Japan). Ccβ-actin (GenBank accession number: OQ658571) was utilized as the reference gene in the qRT-PCR analysis. Specific primer sequences can be found in Appendix A and the melting curve of the qRT-PCR primer for CcFoxO was in Appendix A.

### 4.4. Synthesis of Double-Stranded RNA and RNAi Assays

The production of double-stranded RNAs (dsRNA) targeting EGFP, CcTRPM, and CcFoxO was accomplished using the MEGAscript T7 high yield transcription kit (Cat#AM1334, Thermo Fisher Scientific) with primers (Appendix A) integrated with T7 RNA polymerase promoter sequences. The synthesized dsRNAs were purfied using MEGAclear columns (Ambion) and eluted in nuclease-free water treated with diethyl pyrocarbonate (DEPC). The concentration and purity of the dsRNA were assessed through ultraviolet spectrophotometry and gel electrophoresis, respectively.

Each biological replicate, including at least 30 nymphs, was administered dsRNAs (1000 ng/μL) targeting various genes through a stem-leaf device. Each treatment was repeated three times. For the stem-leaf device, a fresh pear leaf was inserted into a 250 μL PCR tube containing different concentrations of dsRNAs, and the entire assembly was placed in a 50 mL sterile plastic tube. For the RNAi assays, summer-form 1st instar nymphs were segregated into three groups: (1) Nymphs were sampled at 2 and 4 days under 25 °C, while 3 and 6 days under 10 °C for RNAi efficiency analysis via qRT-PCR. (2) Variations in morphological characteristics were monitored every two days under 10 °C by documenting the number of summer-form and winter-form individuals according to the alterations in body color and external characteristics. (3) Samples were collected at 12 days after dsRNA feeding for the assessment of cuticle ultrastructure, total cuticle pigment content analysis, triglyceride and glycogen content determination, and evaluation of cuticle chitin content under 10 °C condition.

### 4.5. Transmission Electron Microscopy (TEM) Analysis of Cuticle Ultrastructure

To compare the nymph cuticle ultrastructure, transmission electron microscopy was applied following established procedures [15,18]. Initially, headless body samples were fixed in 4% polyformaldehyde (PFA) for 48 h, followed by post-fixation in 1% osmium tetroxide for 1.5 h. Subsequently, the samples underwent a standard ethanol/acetone series, followed by infiltration and embedded in spurr resin. Superthin sections (approximately 70 nm) of the thorax were prepared, stained with 5% uranyl acetate and Reynolds’ lead citrate solution. Observation, imaging, and measurements of the sections were performed using a Hitachi HT7800 transmission electron microscope operating at 120 kv. Each sample comprised five nymphs.

### 4.6. Phenotypic Observation of Summer Form and Winter Form

To quantify phenotypic divergence among nymphs under different treatments, three indicators were utilized: total pigment content of the cuticle, cuticle chitin content, and cuticle thickness. The experimental subjects were used 15 days after dsCcFoxO or dsEGFP treatment of 1st-instar nymphs, with 30 nymphs per replicate and 9 replicates for every group.

For the total pigment content analysis, nymph cuticles were dissected and treated with acidified methanol (with 1% concentrated hydrochloric acid). The cuticle tissues were ground, subjected to 24 h of thermostatic oscillator at 200× *g* and 25 °C, and the resulting supernatants were filtered through a 0.45 μm filter membrane to obtain the total pigment extract post centrifugation.

To analyze cuticle chitin content, staining with Triticum vulgaris lectin–fluorescein (WGA-FITC) was performed. Nymph samples were immobilized in 4% PFA for 12 h at 4 °C, dehydrated with sucrose solutions (10%, 20%, and 30%), and rinsed with sterile 1 × PBS solution. Gradual embedding at −25 °C was achieved using SAKURA Tissue-Tek OCT compound. Ultrathin sections (approximately 70 nm) of insect body materials were obtained with a Leica CM1850 cryotome. Staining of cuticle chitin and cell nucleus in the sections were carried out with WGA-FITC (50 μg/mL) and DAPI (10 μg/mL) separately, followed by washing with sterile PBS buffer. Fluorescence images were captured using a Leica SP8 confocal microscope.

To assess the cuticle chitin content, the chitin ELISA kit (Cat# YS80663B, Yaji Biotechnology, China) was employed following the manufacturer’s instructions. Initially, a 1.5 mL sterile centrifuge tube was used to homogenize the sample, consisting of 50 nymphs per sample, using an electric homogenizer, followed by centrifugation at 3000× *g* for 20 min. Subsequently, the samples, along with standard samples, were incubated in ELISA plates for approximately 30 min under 37 °C conditions. The ELISA plates were rigorously washed with a washing buffer at least five times (each time for 30 s). The enzyme labeling reagent was then added to the ELISA plates for further incubation and washing. Finally, the A and B solutions of the chromogenic agent were thoroughly mixed for color development under 37 °C conditions and maintained in the dark.

### 4.7. Analysis of Triglyceride and Glycogen Content

To investigate the energy metabolism of *C. chinensis*, metabolites associated with cold tolerance and energy were measured using commercial kits. The samples were tested using whole insects following the manufacturer’s instructions, encompassing the triglycerides (TG) assay kit (Cat# A110-1-1, Nanjing, Jiancheng, China) and glycogen assay kit (Cat# A043-1-1, Nanjing, Jiancheng, China) [41]. Each treatment consisted of nine biological replicates, with 50 nymphs per replicate.

### 4.8. Nile Red Staining

Summer-form nymphs underwent treatment and were subsequently placed on a glass slide. An adequate volume of PBS buffer was added dropwise to separate the fat bodies at room temperature. The isolated tissue was then transferred to a staining dish filled with PBS buffer. After blotting the PBS buffer, the tissue was fixed with 4% paraformaldehyde (PFA) for 30 min at room temperature. Subsequently, the PFA solution was blotted, and the samples were washed three times with PBS for 10 min each. A solution of 0.5 mg/mL Nile red dye was diluted to a ratio of 1:2500 and applied for overnight staining at 4 °C. After blotting the dye, samples were washed three times with PBS, each wash lasting 10 min. The DAPI mother solution at 20 ng/μL was diluted 1:10 and stained for 5 min. DAPI was then blotted, and a PBS wash was performed three times for 10 min each. Finally, after mounting, the samples were photographed, saved, and scanned using a laser confocal microscope.

### 4.9. Statistical Analysis

IBM SPSS Statistics 26.0 software and GraphPad Prism 8.0 software were employed for data analysis and figure generation, respectively. Results were presented as means ± standard deviation (SD) based on three or nine replicates. Significance among various groups was assessed using Student’s *t*-test (** *p* < 0.01, *** *p* < 0.001, ns: not significant) or one-way analysis of variance (ANOVA) followed by Tukey’s honest significant difference (HSD) test for multiple comparisons (distinguishable letters indicated by *p* < 0.05).

## 5. Conclusions

In summary, we used RNA interference to reveal the regulatory role of a specific gene in the seasonal changes of an insect species. We also looked into how the insect’s response to energy-related substances changed. Our results indicated that under certain temperature conditions, a receptor gene was activated in the insect nymphs, leading to the expression of the gene we studied. This gene then played a role in energy storage, chitin synthesis, and adaptation to cold temperatures, ultimately affecting the insect’s survival rates. These findings not only enhance our understanding of gene function but also offer potential insights for environmentally friendly pest management strategies. Further research is required to fully grasp the mechanisms behind these seasonal changes, including exploring the signaling pathways that control the gene of interest and identifying other genes influenced by its expression.

## Figures and Tables

**Figure 1 ijms-25-08545-f001:**
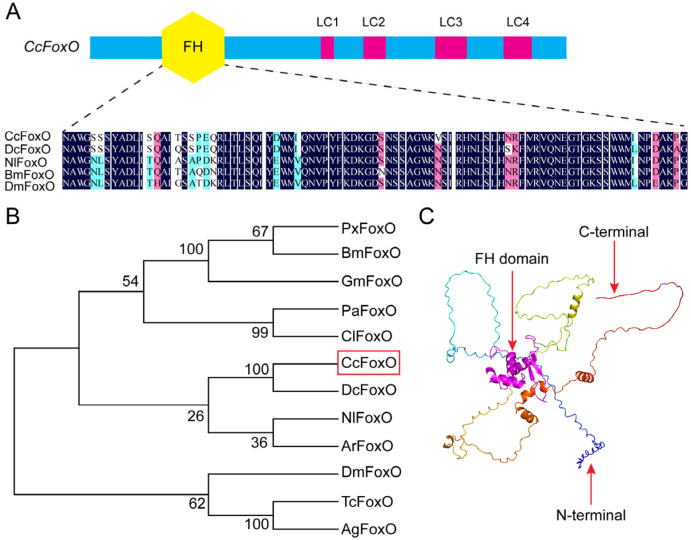
Sequence analysis of FoxO proteins. (**A**) Schematic of *CcFoxO* domains and sequence alignment. *CcFoxO* contains a forkhead (FH) domain. The *CcFoxO* (*C. chinensis*, PP931994) FH domain was aligned with its orthologs from *DcFoxO* (*Diaphorina citri*, KAI5702794.1), *NlFoxO* (*Nilaparvata lugens*, XP_039275858.1), *BmFoxO* (*Bombyx mori*, AFD99125.1), and *DmFoxO* (*Drosophila melanogaster*, NP_996205.1). The same gene name and GenBank accession number are as below. (**B**) Phylogenetic analysis of FoxO orthologs from 12 insect species based on the amino acid sequences. The phylogenetic tree (bootstraps with 1000 replicates) was constructed by MEGA-X using maximum-likelihood methods. The *CcFoxO* protein was set as an outgroup control. *PxFoxO*: (*Plutella xylostella*, XP_037964391.1); *GmFoxO*: (*Grapholita molesta*, QIM56595.1); *PaFoxO*: (*Pyrrhocoris apterus*, XBC28246.1); *ClFoxO*: (*Cimex lectularius*, XP_014254467.1); *ArFoxO*: (*Athalia rosae*, XP_048514911.1); *TcFoxO*: (*Tribolium castaneum*, XP_975200.2); *AgFoxO*: (*Anoplophora glabripennis*, XP_018574886.1). (**C**) The predicted protein tertiary structure of *CcFoxO*. The conserved forkhead (FH) domain was indicated in the center of the three-dimensional structure.

**Figure 2 ijms-25-08545-f002:**
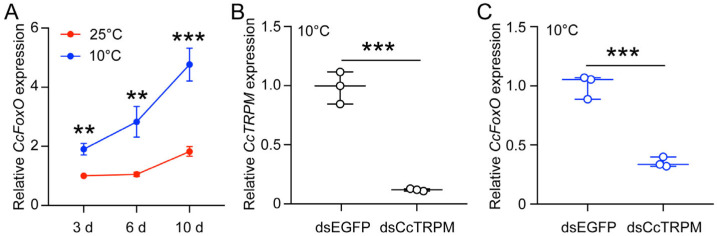
Expression pattern analysis of *CcFoxO*. (**A**) The mRNA expression of *CcFoxO* in SF first instar nymphs in response to different temperatures at day 3, 6 and 10 of 25 °C and 10 °C by qRT-PCR (*n* = 3). Expression level of *CcFoxO* at day 3 under 25 °C is used as 1. (**B**) RNAi efficiency of dsCcTRPM after treatment at 48 h compared to dsEGPF feeding (*n* = 3). Expression level of *CcTRPM* under dsCcTRPM treatment is used as 1. (**C**) Effect of *CcTRPM* knockdown on the mRNA expression of *CcFoxO* (*n* = 3). Expression level of *CcFoxO* under dsCcTRPM treatment is used as 1. Newly hatched summer-form nymphs were worked on as the experimental subjects. The results are indicated as the mean ± Standard Deviation (SD) and obtained from three independent biological replicates, with per replicate consisting of at least 30 nymphs. Statistical significance was determined using the pairwise Student’s *t*-test, and significance levels are shown as ** (*p* < 0.01) or *** (*p* < 0.001).

**Figure 3 ijms-25-08545-f003:**
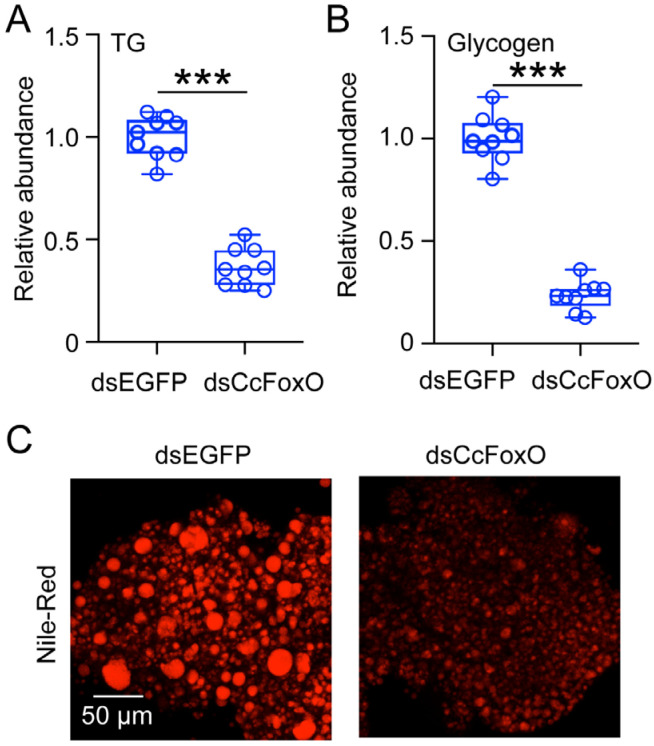
Effect of *CcFoxO* knockdown on the abundance of triglycerides, glycogen, and lipid droplet. (**A**,**B**) Comparison of the nymph triglycerides and glycogen content after treatment with dsEGFP and dsCcFoxO at 48 h (*n* = 9). (**C**) Lipid droplets stained with Nile red in fat bodies of SF first instar nymphs after being treated with dsEGFP and dsCcFoxO at 48 h. Representative images were captured, where lipid droplets are shown in red. The scale bar represents 50 μm. Statistical significance was determined using the pairwise Student’s *t*-test, and significance levels are shown as *** (*p* < 0.001).

**Figure 4 ijms-25-08545-f004:**
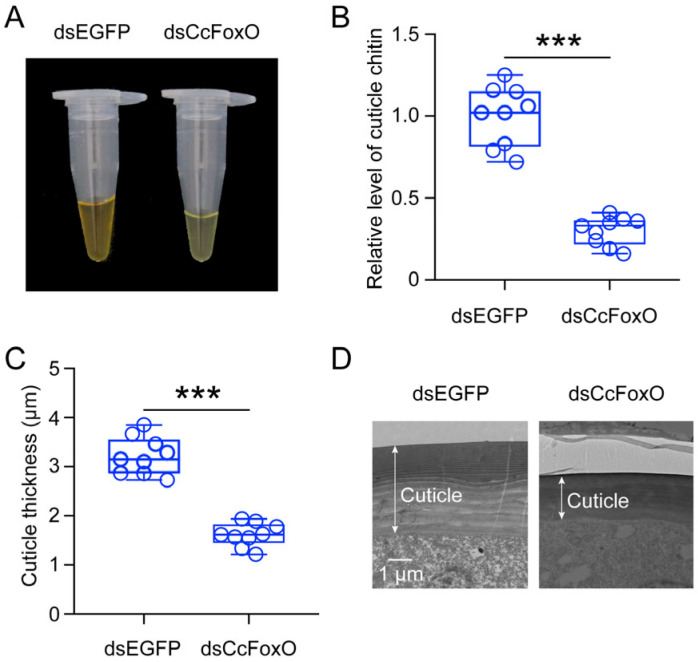
Comparison of the total cuticle pigment content (**A**), cuticle chitin content (**B**), and cuticle thickness (**C**,**D**) of SF first instar nymphs treated with dsCcFoxO and dsEGFP at 15 days. Data in (**B**,**C**) were presented as means ± SD with nine independent biological replicates, and each circle indicated one biological replicate. Scale bar in (**D**) is 1 μm. Student’s *t*-test of the pairwise was used to do the statistically analysis (*** *p* < 0.001).

**Figure 5 ijms-25-08545-f005:**
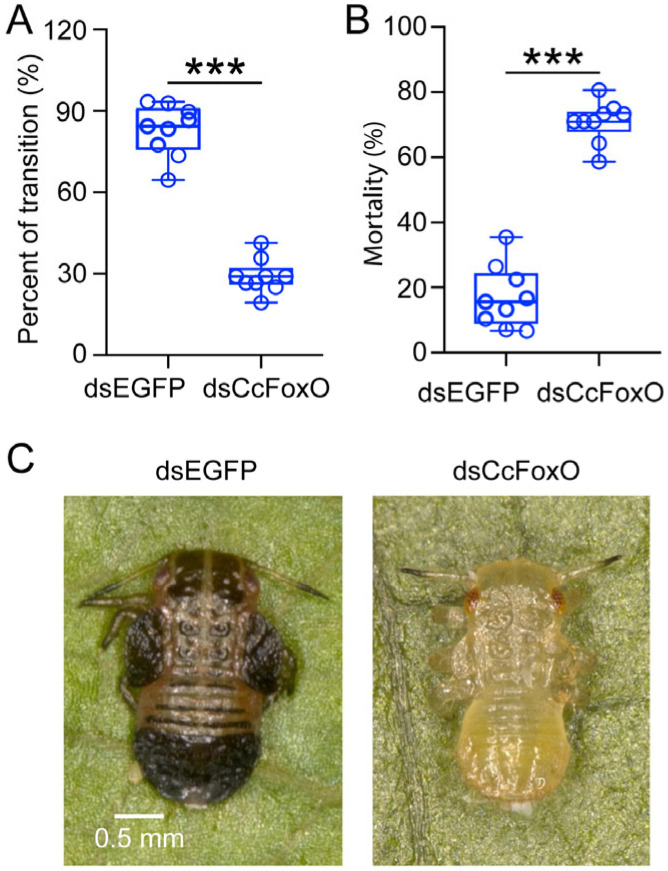
Effect of *CcFoxO* knockdown on the transition from summer form to winter form and mortality. (**A**,**B**) The transition percent and mortality of SF first instar nymphs treated with dsEGFP or dsCcFoxO at 15 days under 10 °C (*n* = 9). (**C**) The phenotypes of SF first instar nymphs treated with dsEGFP and dsFoxO at 15 day under 10 °C. Scale bar is 0.5 mm. Data in (**A**,**B**) are presented as mean ± SD with nine biological replications. Statistically significant differences were determined with the pair-wise Student’s *t*-test, and significance levels were denoted by *** (*p* < 0.001).

**Figure 6 ijms-25-08545-f006:**
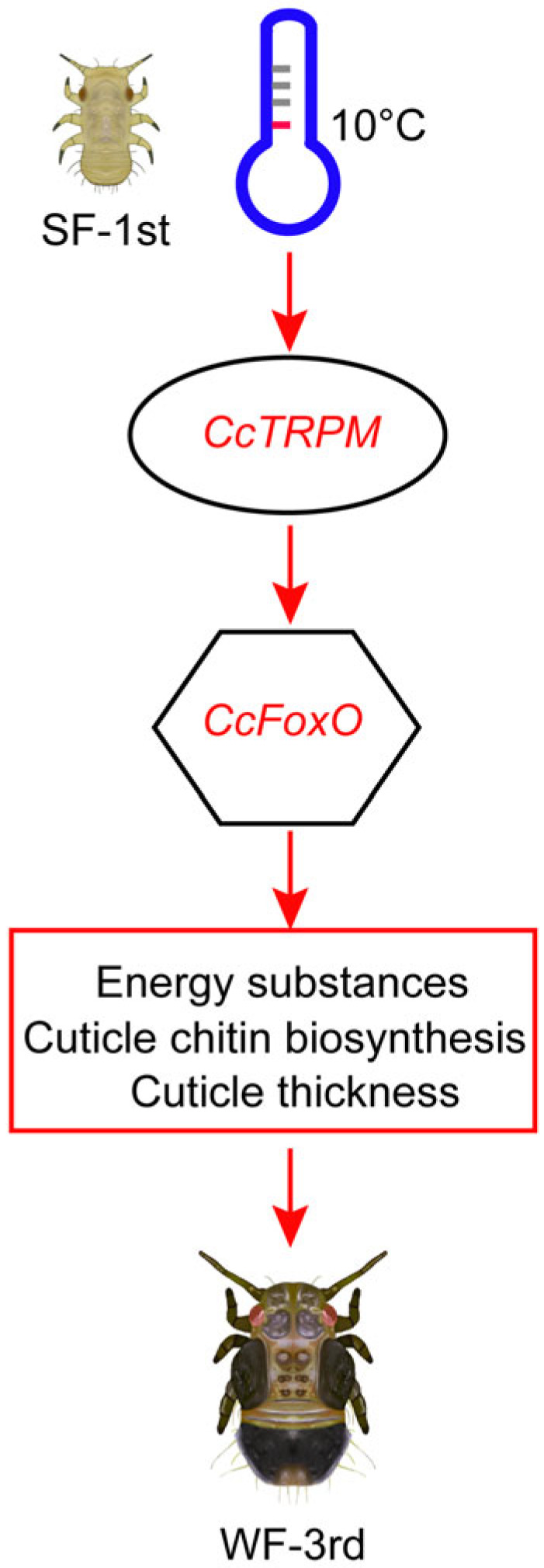
A model showing the key roles of transcription factor *CcFoxO* in the transition from summer form to winter form. Under 10 °C condition, low temperatures induced the activation of the temperature receptor *CcTRPM* in SF first nymphs. Then, *CcTRPM* significantly increased the expression of transcription factor *CcFoxO*, promoting the accumulation of energy substances, rising cuticle chitin biosynthesis, as well as cuticle thickness. As a result, the first instar nymphs of SF developed to third instar nymphs of WF in order to better adapt to low temperatures.

## Data Availability

All the data are in this manuscript.

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
