# Peer review of "Transcription Factor CcFoxO Mediated the Transition from Summer Form to Winter Form in Cacopsylla chinensis"

_ijms, 2024, doi:10.3390/ijms25158545_

Round 1

Reviewer 1 Report

Comments and Suggestions for Authors

Cacopsylla chinensis displays distinct summer-form and winter-form distinguished by significant morphological variations. Previous studies have highlighted the role of temperature receptor CcTPRM in orchestrating the transition in response to 10 °C temperature, but the detailed downstream signals remain unclear. In this manuscript, Wei and Zhang identified a transcription factor CcFoxO and investigated its role in the seasonal polyphenism of C. chinensis. RNAi-mediated knockdown revealed that CcFoxO facilitates the accumulation of triglycerides and glycogen, thereby influencing the transition from summer-form to winter-form by affecting cuticle pigment content, cuticle chitin levels, and cuticle thickness. These insights not only enhance our comprehension of FoxO functionality but also offer avenues for environmentally friendly management strategies for C. chinensis. However, some grammar needs to be revised and some additional clarification may strengthen the manuscript.

Specific suggestions

1. In Line 16, change “transcription factor CcFoxO” to “C. chinensis FoxO (CcFoxO)”. 

2. In Line 19, change “was” to “is” and keep consistent in tense.

3. In Line 53, change “assumes” to “assume”.

4. In Line 64, change “larvae” to “nymphs”.

5. In Line 59, 101-102, the abbreviations should appear for the first time, so JH and 20E should be added in Line 59, then deleted the later.

6. In Figure 2A, how to prove the primer specificity for CcFoxO quantification? Added the melting curves for qRT-PCR primers?

7. In Line 367, which tissue was used for transmission electron microscopy?

8. In Line 321-323, how did the authors search the protein sequences? Was it taxonomically restricted? These details would be much appreciated.

9. Please supply the detail protocol for the Nile red staining.

10. In Line 413, “t” in “t-test” should be italic.

11. In References, deleted “PLOS Computational Biology/” in Line 469 and “/National Science Review” in Line 472; delete the blank space after “control” in Line 476; delete the first word “in” in Line 488 and 501.

Comments on the Quality of English Language

Line 19, change "was" to "is"

Line 53, change "assumes" to "assume"

Line 64, change "larvae" to "nymphs"

Reviewer 2 Report

Comments and Suggestions for Authors

Dear authors and editor,

The manuscript by Wei et al. reports a gene function study of transcription factor CcFoxO in mediating the transition from summer-form to winter-form in Cacopsylla chinensis using RNAi. The authors found the low expression of CcFoxO under low temperatures 10 °C and control temperature. This gene might be regulated by CcTRPM, a temperature receptor in orchestrating the transition in response to low temperature. The results showed that after decreasing the expression of CcFoxO by feeding dsRNA, it led to the substantial decreased accumulation of triglycerides and glycogen which was related to energy metabolism. And it could significantly reduce cuticle pigment content, cuticle chitin levels, and cuticle thickness. These factors contributed to the failed transition and high mortality. Overall, this study performed a good experimental design to test the CcFoxO function in non-model species. And it could give us more information about the key gene function involved in transition and might provide potential target genes for pest control.  However, I have some concerns for the authors that needs to be addressed.

Firstly, some important references are missing in the introduction and discussion part such as line 31, 33, 35, 38, 43, 52, 64, 66, 74, 78, 89, 103, 111, 236-237,238, 244, 283, 285. It would be necessary to include the references to let the readers have access to these papers.

I am a bit confused about the RNAi for dsCcTRPM. The authors mentioned this could show the efficiency for the RNAi and CcFoxO acts downstream of CcTRPM. But as the paper mainly focused on the CcFoxO, I would like the authors to stress more why they did both gene RNAi. I think I could find this information in the abstract, but it would be good to mention this study again in the introduction part. Furthermore, would the authors plan to also check the cuticle for dsCcTRPM treatments? And what is the expression of CcTRPM in dsCcFoxO treatment? It would be good to see the expression of CcTRPM at 25 and 10C at different time points.

Figure 2 A-C: be specific about what is used as 1 in the gene expression. For Figure B and C, Which days of nymphs have been used in the results? It would be great to also include this in the figure legend to keep the reader easy to follow. And the same for other figures.

Line 217-226: it is a very interesting finding for the gene function. What was the mortality of dsCcFox transition individuals? Were they all dead? It would be good to add this information to the result section. And I assume this gene might be important for other function as the non-transited individuals also with a high mortality. 

Discussion part (line 252-290). It would be good if the author could discuss the regulating glycogen and triglycerides first and then decrease in the transition as this would be easy to follow and understand logically. Could line 276-284 also be the reason also for high mortality in dsCcFoxO?  

It would be clear if the authors could add more information such as the replicate number and treatments into method section. And some information mentioned in the result section were the information needed for the method section (line 180-181, line 199-200). The order of result section should match to the method section, or it would be hard to follow.

For example:

Line 329: why do the authors only use the summer-form nymphs?

Line 330: how to expose to temperature 10 and 25? Is it in the incubators?

Line 344: what are the primers? 

Line 354-355: be specific why the authors chose these days. Maybe because they would be 2nd instar nymphs??

Line 351-352: how often the dsRNA has been changed at different temperature?

Line 352: be specific about the concentrations of dsRNAs.

Line 373: be specific about the treatment which I might think it is mentioned in line 218?? 

Line 376-400: give information about how many nymphs or groups were used for total pigment content of the cuticle, cuticle chitin content, and cuticle thickness.

Line 401-406: give information about how many nymphs were used.

Line 9-11: combine these two sentences as the background part is too long in the abstract.

Line 13-14: stress this information in the introduction part.

Line 16-17: add “transition from summer to winter-form under cold stress by using RNAi” 

Line 21: it would be important also mention the decreased gene expression in CcFoxO led to high mortality and failed transition.

Line 39-40: separate the influence effects into biotic and abiotic factors.

Line 44: rephrase the sentence to “human can better understand how to manage agricultural population under certain environmental conditions”.

Line 46: add “Despite the importance of polyphenism, contemporary investigation primarily center on model insects.”

Line 47: change “certain aphids” to “pea aphids” 

Line 52: add the species “Pyrrhocoris apterus”

Line 62: rephrase “However, research on multiple transcription factors that ultimately regulate polyphenism in non-model species responses remains scarce and unclear.” 

Line 64: could be more specific about the ‘a significant threat’ such as adding the loss 

Line 71-73: Could summer and winter form be transited if the temperature change?

Line 74-76: It might be clear to add the temperature range for “winter and summer condition” to show the temperature tolerance difference in these two forms.

Line 80: change “of” to “around”

Line 88-100: could be simplify and combine with the following paragraph to stress the DNA-binding domain and the function in insects.

Line 103: change “more” to “other key physiological processes in insects”.

Line 111-113: rephrase the sentence to “Consequently, CcFoxO might play a significant role in the shift from summer-form to winter-form in C. chinensis, which further elucidation is warranted.”

Line 116: could the authors explain why they also targeted CcTRPM? What is the relationship in previous study about CcFoxO and CcTRPM? It would be good to move the information from line 149-151 to introduction.

Line 151-154: move this to the discussion part.

Line 171: add the time points. Why did the authors choose these time points? This information was also missing in the methods part.

Line 183-185: for the nile-red strain results. Is the decrease based on both the lightness and size?

Line 253-254: rephrase this sentence as the authors need to add the result of CcTRPM at 25 and 10C to show the similar pattern under low temperature. 

Line 256-257: it would be clear if the authors could stress the importance for the relationship between CcFoxO and CcTRPM.

Line 293-296: move this information ahead to show previous study.

Table S1: is there any primer from other study? If so, please add the reference. And also refer to certain primers’ name in the methods and refer to this table.

Comments on the Quality of English Language

Overall it is fine but could be improved by some modification.

Reviewer 3 Report

Comments and Suggestions for Authors

The study entitled “Transcription factor CcFoxO mediated the transition from summer-form to winter-form in Cacopsylla chinensis” investigates the correlation between the transcription factor CcFoxO and low temperatures. The authors report that CcFoxO is involved in the transitioning process from summer-form to winter-form through the regulation of the cuticle pigmentation and thickness. The manuscript is overall very well written and provides valuable insights for the role of CcFoxO in seasonal polyphenism that can be used from pest control programs. I only have a few comments and I suggest it for publication after minor revision.

In general the introduction is well written although some of the sentences are quite long and some of the wording is very sophisticated. I would suggest avoiding long sentences for better clarity and try to use more commonly used terms. For example, “evolutionary victors” and “innate wisdom" could be replaced, as they are too poetic. In addition the use of temperature units is not consistent (10oC vs 10 oC).

L43:  "Leveraging this mechanism of adaptive response, humans can effectively manipulate environmental conditions to manage agricultural pest populations." Authors need to elaborate on this statement and make this point stronger.

L141-142: provide p values to support this statement

L217-218: provide p values to support this statement

L244-247: what is the ecological significance of polyphenism in these insects?

L254-257: how are these findings related to ecological strategies of C. chinensis?

L414: conclusions are very narrow. I suggest authors to rewrite this part and make it more general and avoid references to figures and specific experiments they conducted.

Comments on the Quality of English Language

Minor revision required

Round 2

Reviewer 2 Report

Comments and Suggestions for Authors

Dear authors and editor,

The authors have thoroughly responded to all concerns I brought up in review and all prior comments and suggested edits have been addressed appropriately.